# Hunger, Satiety, and Their Vulnerabilities

**DOI:** 10.3390/nu16173013

**Published:** 2024-09-06

**Authors:** Richard J. Stevenson, Kerri Boutelle

**Affiliations:** 1School of Psychology, Macquarie University, Sydney, NSW 2109, Australia; 2Department of Pediatrics, Herbert Wertheim School of Public Health and Human Longevity Science and Psychiatry, University of California San Diego, San Diego, CA 92161, USA; kboutelle@health.ucsd.edu

**Keywords:** hunger, satiety, interoception, temporal cues, medial temporal lobe, remediation, appetite, declarative memory

## Abstract

The psychological states of hunger and satiety play an important role in regulating human food intake. Several lines of evidence suggest that these states rely upon declarative learning and memory processes, which are based primarily in the medial temporal lobes (MTL). The MTL, and particularly the hippocampus, is unusual in that it is especially vulnerable to insult. Consequently, we examine here the impact on hunger and satiety of conditions that: (1) are central to ingestive behaviour and where there is evidence of MTL pathology (i.e., habitual consumption of a Western-style diet, obesity, and anorexia nervosa); and (2) where there is overwhelming evidence of MTL pathology, but where ingestive behaviour is not thought central (i.e., temporal lobe epilepsy and post-traumatic stress disorder). While for some of these conditions the evidence base is currently limited, the general conclusion is that MTL impairment is linked, sometimes strongly, to dysfunctional hunger and satiety. This focus on the MTL, and declarative learning and memory processes, has implications for the development of alternative treatment approaches for the regulation of appetite.

## 1. Introduction

The psychological states of hunger—the desire to eat—and satiety—the feeling of having eaten—play a key role in regulating food intake (e.g., [1,2,3,4,5]). A growing body of research suggests that learning and memory processes are central to understanding these psychological states (e.g., [6,7,8,9,10,11]). Human learning and memory are generally conceived as a set of discrete systems (e.g., [12,13,14]). Of these, the declarative learning and memory system [15,16] is most relevant here, as it appears instrumental in supporting hunger and satiety [7,8]. The primary neural substrates of declarative learning and memory are the medial temporal lobes (MTL; [17,18,19]), and to a lesser extent, the anterior temporal lobes (ATL; [20,21]). It has been known for many years that surgical ablation of the MTL, notably for intractable epilepsy, can seriously impair hunger and satiety, and appetite regulation more broadly (e.g., [22,23]). However, what is not so well appreciated, is that the MTL, and particularly the hippocampus, can be impaired by a large range of insults (e.g., [24,25,26,27,28]). Many of these occur with high frequency in the population (e.g., adiposity, type 2 diabetes, stress, depression, etc.). In this manuscript, the aim is to document what is known about hunger and satiety in: (1) situations/conditions that are strongly linked to ingestive behaviour and where these are evidence of MTL pathology (i.e., a Western-style diet, obesity, and anorexia nervosa); and (2) conditions that primarily involve MTL pathology, but where ingestive behaviour is not central (i.e., temporal lobe epilepsy and post-traumatic stress disorder). The final section of the manuscript examines the implication of these findings for optimising hunger and satiety in these and related conditions.

## 2. Hunger and Satiety, Learning and Memory, and the Medial Temporal Lobes (MTL)

Several strands of evidence suggest that learning and memory are central for understanding the psychological basis of hunger and satiety (e.g., [6,7,8,9,10,11]). One compelling finding comes from the study of brain-injured people, and particularly the case of Henry Molaison (HM). HM underwent a bilateral temporal lobectomy for intractable epilepsy, significantly impairing each component of his declarative learning and memory system—episodic memory (i.e., personal events, such as what one had for breakfast) and semantic memory (i.e., factual knowledge, such as the meaning of a red traffic light).

When the full extent of HM’s deficits were realised, he was the subject of intense scientific interest for the remaining 40 years of his life. Sometimes, his testing sessions went well over scheduled mealtimes, and it was observed that he never complained of hunger [29]. More formal investigation followed and involved HM evaluating his internal state before and after meals for 6 days on a 100-point scale from very hungry to very full. His ratings changed little, being around 50 irrespective of when they were obtained, and markedly differed from a control sample of normal and brain-injured controls (*n* = 8), who all reported increased fullness after a meal [22]. These observations were supported by another striking finding, one that was not immediately apparent because of his highly structured life in residential care. HM was offered a second main meal almost immediately after eating his first [22]. He readily consumed this second main meal, all except the salad. This was the only occasion where HM reported any change in state after a meal, with his rating shifting from 50 to 75. 

HM is not unique. Eating multiple meals was documented in several other patients with MTL damage [23,30]. Similarly, significant abnormalities in hunger and satiety were also found in these and other similar cases [23,30,31,32]. The implication of these neuropsychological findings is that relatively selective damage to brain areas that support declarative learning and memory cause serious or even catastrophic impairment to the experience of hunger and satiety.

In Cannon and Washburn’s [33] pioneering study of hunger, exploring its basis in stomach contractions, they identified associative learning as one additional means of generating hunger. With reference to Pavlov, they suggested that environmental events that are predictive of rewarding food, namely those that taste good and are high in calories, result in an appetite or hunger for that food. This idea saw widespread adoption and theoretical elaboration in the century since. There are several contemporary models (e.g., [34,35,36]), and all envisage a hunger process similar to that depicted in panel A of Figure 1. Here, an environmental cue such as the appearance of a food, its smell, packaging, advertising jingle, logo, etc., can cause episodic memory retrieval of eating that food [7,8]. Hunger occurs when episodic retrieval takes place, as people briefly re-experience the affective properties of that specific food (or class of food). This affective experience is one form of hunger, and at its most intense it can constitute a food craving (noting that the earliest of these contemporary models [35] was originally envisaged to explain food cravings). Both episodic memory and Pavlovian learning draw upon structures in the MTL, with the former reliant on the hippocampal formation (e.g., [37,38]) and the latter reliant upon the amygdala (i.e., within the MTL) and striatum (outside the MTL; e.g., [39]). This reliance on MTL-based structures is why HM had such significant impairments in hunger.

It is widely held that people also experience another form of hunger, namely a general desire (i.e., general hunger) for any food (e.g., [33,35,36,40,41]. This too appears to have a Pavlovian basis and two types of cues are important here. One concerns various bodily sensations generated by the physiological changes that accompany digestion (e.g., an empty rumbling stomach) and energy-usage (e.g., fatigue) that predict that food will be rewarding to eat now—interoceptive hunger cues (e.g., [42,43,44]). The other is temporal cues, such as knowing it is lunchtime or that several hours elapsed since the last meal (e.g., [9,45]). The meaning of bodily and temporal hunger cues are probably learned, with this occurring during childhood [46,47,48]. The nature of this learning process is different to that which occurs when a person associates an environmental cue with a particular palatable food. During development, interoceptive and temporal cues can occur with many hundreds of different food types [7,8]. The learning here produces general knowledge. This may manifest in two ways, as illustrated in panel B of Figure 1. First, the temporal or interoceptive cue acquires an explicit meaning—semantic knowledge—that any food will be rewarding to eat now (i.e., general hunger). Second, the semantic memory also consists of an averaged record of those previously enjoyed eating bouts that took place with or after the temporal or interoceptive cue. This may be used to affectively augment the memory of any subsequently encountered food cue, thereby increasing the expected pleasure arising from it (i.e., boosting hunger)—a form of excitation transfer where affect/arousal from one cognitive process is transferred to another (e.g., [49,50]). The neural basis of these hypothesised processes again resides in the MTL, and because of semantic memory involvement, the ATL as well (e.g., [51,52,53,54,55]). A few studies reported that damage to the MTL and/or ATL impairs the capacity of interoceptive and temporal hunger cues to generate hunger (e.g., [56,57]). The reliance of general hunger on MTL structures likely further contributed to HM’s deficits.

The final aspects of this declarative learning and memory account pertains to satiety and are illustrated in panel C of Figure 1. The animal literature is particularly important in developing accounts in this field, especially the notion of memory inhibition—the cognitive process of stopping certain memories being retrieved [58,59,60,61,62,63]. In much the same way that an interoceptive cue can come to signal that food will be rewarding now, other interoceptive cues associated with satiation and satiety, can come to predict that food will not be rewarding now [6]. Not only does this have a declarative component (i.e., semantic knowledge—I have eaten, I am full), there are also effects on memory processing. These effects involve memory inhibition, with this process driven both by satiety signals [58,59,60] and by physiological markers of long-term energy stores [7]. 

As time elapses from completion of a meal, satiation and satiety cues wane and so does memory inhibition, but never entirely. All memory inhibition is only lost when markers of long-term energy stores are depleted or where there is neural damage. Under these conditions, any cue remotely linked to food can bring thoughts of food to mind (e.g., a cloud reminds one of a loaf of bread). In more typical circumstances, memory inhibition serves to dampen both the ability of Pavlovian cues to bring thoughts of food to mind and the retrieval of pleasant food-related memories [64,65]. The neural basis of memory inhibition was studied extensively in animals, and somewhat in people, and again it depends upon MTL structures (e.g., [60,65,66,67]). Not surprisingly then, HM also had deficits in satiety, having no episodic memory of his recent eating bouts, impaired semantic knowledge of what his interoceptive satiety cues meant, and little capacity to enact memory inhibition. 

The declarative learning and memory model of hunger outlined here draws upon much prior work and depends to a significant degree on an intact MTL and its adjacent structures (e.g., [6,7,8,35,36,40,68,69,70]). While discrete lesions to these structures are relatively rare—hence interest in cases such as HM—the MTL and particularly the hippocampal formation, are especially vulnerable to insult (e.g., [24,25,26,27,28]). If declarative learning and memory processes are important for supporting hunger and satiety, this would imply that MTL impairments should be accompanied by their dysfunction.

## 3. Vulnerabilities of Hunger and Satiety

The subsequent sections examine evidence for impaired hunger and satiety in conditions that were linked to MTL dysfunction, and especially of the hippocampal formation. We label this as ‘vulnerabilities of Hunger and Satiety’ primarily because it appears that the MTL, and especially the hippocampus, is particularly easy to damage as we described earlier and as discussed further towards the end of the manuscript (e.g., [24,25,26,27,28]). To the extent then that hunger and satiety rely on MTL-dependent processing this should make them both also especially vulnerable to impairment.

We focus here on: (1) situations/conditions that are closely connected with eating and where there is evidence of MTL pathology; and (2) conditions not generally connected with eating but where evidence for MTL pathology is overwhelming. For the former, we examine Western-style diet, obesity, and anorexia nervosa. For the latter, we focus on temporal lobe epilepsy and post-traumatic stress disorder (PTSD).

## 4. Western-Style Diet 

A Western-style diet (WS-diet) is one rich in added sugar, saturated fat, and salt, and is a common dietary pattern in many developed nations and one composed of many ultra-processed foods (e.g., [71]). A WS-diet impairs MTL function—especially the hippocampal formation—and does so long before any excess weight gain takes place. This claim is supported both by a large body of animal research [72] and a growing body of human findings [73]. The latter includes human experimental studies [65,74], which indicate that exposure to a WS-diet produces a rapid and selective impairment on neuropsychological tasks that measure hippocampal-dependent learning and memory (HDLM).

Evidence of dysfunctional hunger and satiety in habitual consumers of a WS-diet came from two main sources. One is to examine self-reports of hunger and satiety before and after a meal in lean healthy people who differ in their habitual diet. For example, Francis and Stevenson [75], identified participants who differed in their adherence to a WS-diet using a food frequency measure [76]. They selected those most divergent in this regard (two groups of *n* = 16) and evaluated hunger and satiety ratings before and after a snack and main meal presented in the laboratory. Participants who consumed a WS-diet ate significantly more food at both meals (M total kJ = 3171) than those with a healthier diet (M total kJ = 2158). However, both groups showed almost identical changes in their ratings of hunger and satiety. Importantly, this suggests that greater quantities of food are needed to shift interoceptive ratings in consumers of a WS-diet, implying an interoceptive blunting. 

Of course, it is not possible to know from these data if such blunting preceded consumption of a WS-diet or was a consequence of it. One approach to this question was to randomly assign lean students who normally consume a healthier diet to five days of a WS-diet [74]. Here, participants were randomised to either five successive days of a breakfast composed of a large quantity of sugar and saturated fat (WS-diet group, *n* = 50; 30% saturated fat; and 18% added sugar) or to a healthy breakfast (control group, *n* = 50; 6% saturated fat, and 11% added sugar) broadly matched for calories, palatability, and volume. These breakfasts were eaten in the laboratory each morning, ensuring the diet manipulation took place, with additional food intake monitored by diet diaries. In those exposed to the WS-diet, smaller changes in hunger and satiety across a test breakfast were observed at the end of the intervention period relative to the beginning when compared to controls maintained on a healthier diet [74]. This suggests that the observed blunting of hunger and satiety are a consequence of a WS-diet.

While this blunting of hunger and satiety is reminiscent of that seen in people with MTL lesions, it is not yet possible to tell if more specific deficits might be present. Hunger ratings do not allow for differentiation between the various processes illustrated in Figure 1. When participants make a hunger rating, they may be having thoughts/images about a specific food (i.e., food-specific hunger), they may be trying to attend to their stomach to see if it is empty (i.e., interoceptive cue), or they might be checking to see if it is lunchtime (i.e., temporal cue), or some combination thereof. So, while it is evident that some blunting of hunger is taking place, it is not clear where this is happening. 

For satiety, somewhat more is known, based upon the following experimental approach [65,74]. Before lunch, hungry participants are asked to judge using a line-rating scale how much they want to eat particular snack foods, which they are shown (i.e., food-specific hunger). They then consume a small morsel of each snack food and rate how much they like it (i.e., sensory pleasure/consummatory affect). After a filling lunch, the same tests are repeated. A key finding is that food-specific hunger declines far more across a meal than liking ratings of the snack foods when eaten (see Figure 2). This effect is particularly robust and was replicated several times, with a large average effect size (i.e., Cohen’s d > 0.8; [70,77]).

Participants who consume a healthy dietary pattern (see upper panel Figure 2) have a much larger discrepancy (termed the affective discrepancy effect) between food-specific hunger and liking when they are sated, than those who consume a WS-diet (see lower panel of Figure 2; [65,66]). This effect was also demonstrated cross-sectionally [66] and experimentally [65]. Cross-sectionally [66], participants (*n* = 97) were again identified using the same food frequency measure [75] so as to contrast those with the greatest and least adherence to a WS-diet. The greater the adherence to a WS-diet, the smaller the affective discrepancy effect when tested using the paradigm described above (i.e., wanting and liking ratings for palatable snacks before and after a meal) and with a regression approach. An identical effect was also demonstrated experimentally [65] by randomising participants who normally consume a healthy diet to either five mornings of a breakfast rich in added sugar and saturated fat (WS-diet group, *n* = 55; 19% saturated fat, and 29% sugar) or a control breakfast (control group, *n* = 55; 5% saturated fat, and 10% sugar), broadly matched for energy content, palatability, and volume [65]. In this case, participants in the WS-diet condition showed a smaller affective discrepancy effect after the five-day intervention relative to controls.

Both of these studies also included neuropsychological tests to assess HDLM (word list learning), which was found to be significantly poorer in the cross-sectional study in those with a WS-diet [66] and became poorer in those exposed to a WS-diet in the experimental study [65]. Much of the group differences in the affective discrepancy effect were eliminated if performance on tests of HDLM were taken into account (i.e., statistically, using regression). This suggests two conclusions. First, it implies an MTL substrate for the difference in the affective discrepancy effect between people who consume a WS-diet and those with a healthy dietary pattern. Second, it suggests that when a person who regularly consumes a WS-diet looks at palatable snack foods when they are sated, these foods are more hunger evoking—appealing and tempting that is—than in consumers of a healthier diet. This reflects an impairment in satiety, where it is presumed that memory inhibition mediated by the MTL is no longer so effective, hence allowing the continued retrieval of pleasant food-related memories [6,65].

One major issue that remains unresolved, is whether WS-diet-induced MTL impairment equally or differentially affects food-specific and general hunger. For food-specific hunger, looking at the component processes involved in generating it (i.e., the two green arrows in Figure 1A), selective hippocampal lesions do not seem to impair the capacity for simple Pavlovian learning such as that between an environmental cue and a food—at least in rodents (e.g., [55,78]). This would suggest that the first component of food-specific hunger should be intact (i.e., see Figure 1A, left-hand green arrow). For the second component involving episodic retrieval (i.e., see Figure 1A, right-hand green arrow), some impairment might be expected. The basis for this claim rests on both poorer performance on neuropsychological tests of HDLM [73] and of increased memory failures [79] observed in habitual consumers of a WS-diet. However, as episodic memory retrieval is often for highly palatable foods in consumers of a WS-diet, any impact on retrieval may be relatively modest. This is because of the robust memories for these foods that results from their high reward value (e.g., [80]). Overall, WS-diet-induced MTL impairment may have some limited negative impact on specific hunger for palatable food.

For general hunger, two considerations suggest that WS-diet-induced MTL impairment may have a more significant impact. The first concerns the process of excitation transfer (e.g., [49,50]). This process, described earlier, concerns the transfer of affective arousal from a semantic memory cued by an interoceptive or temporal cue, to an episodic food memory (see Figure 1B). The transfer of affect can occur for any food, including those of low-to-moderate palatability (i.e., many fruits and vegetables). However, episodic retrieval of memories of low-to-moderate-palatability foods (i.e., of lesser reward value) may be more impaired by MTL damage than that for highly palatable foods (i.e., of greater reward value). This is because low-to-moderate reward value foods may have less robust associations. This would then impact the ability of excitation transfer to augment the imagined pleasure of eating low-to-moderate-palatability foods. 

There is a second consideration as well. Hippocampal lesions in rodents affect the learning process that purportedly supports excitation transfer (e.g., [54,55]). This would suggest WS-diet-induced MTL impairment might impair the capacity for excitation transfer. Together, these putative effects would mean that when a general hunger cue is experienced (i.e., a temporal [e.g., its lunchtime] or interoceptive cue [e.g., a rumbling stomach]) encountering low-to-moderate-palatability foods would not generate much hunger. If WS-diet-induced MTL impairment has more severe consequences for general hunger than for specific hunger, then this would be problematic. It would leave a person with food-specific hunger for highly palatable foods, with this process being unconstrained by satiety due to dysfunctional memory inhibition. 

There is also a further consequence. Exposure to a diet that contains a lot of highly palatable foods may have additional adverse effects that operate via memory but independent of any impact on the MTL. This concerns both judgmental contexts (e.g., [81]), and the ‘mental landscape’ of food memories established by an individual (e.g., [82,83]. A frequent consumer of a WS-diet is likely to learn a lot of environmental-cue-ultra-processed food associations, as such foods are heavily advertised. Encountering these advertisements, which is frequent, would then lead a habitual consumer of a WS-diet to experience specific hunger far more often than someone with a healthier dietary pattern. In addition, the experience of frequent positive affect from food-specific hunger may serve as an unflattering frame of reference for imagining the taste of foods such as fruit and vegetables. This would in turn make fruits and vegetables seem even less appealing. 

In sum, the picture that emerges for a WS-diet consumer is of a person: (1) whose food-specific hunger is frequently activated by palatable food cues in the environment; (2) who experiences reduced general hunger with less desire for low-to-moderate-palatability foods; and (3) for whom satiety does not blunt the rewarding properties of palatable food cues. In other words, a person who often exhibits hunger for highly palatable foods when such cues are encountered (which they will be frequently due to advertising), but who is rarely hungry for food in general.

## 5. Obesity

Neuropsychological testing indicates several types of cognitive deficit in people living with obesity, including MTL-dependent learning and memory functions [84]. Neuroimaging also reveals a relationship between adiposity and indices of MTL integrity, when controlling for confounding variables such as diabetes [85,86]. Some of these MTL deficits are probably a consequence of adiposity and some may contribute to it (e.g., [87,88]). Irrespective of their cause, the presence of MTL impairments would, from the perspective of this manuscript, suggest abnormalities in hunger and satiety. 

Interest in dysfunctional hunger and satiety in obesity has a long history. In the 1960s, it was hypothesised that people with obesity were less sensitive to interoceptive cues and more sensitive to environmental ones [89]. As a result, when they encountered environmental food cues, they experienced a strong hunger to consume that food. This enhanced food cue reactivity was further hypothesised to be combined with another impairment: reduced capacity to experience interoceptive cues signaling either hunger or satiety. The hypothetical consequence of all of this would be weight gain and obesity. 

In the years since this theory was outlined, both of its core elements retained appeal. Indeed, they contributed to the development of Wardle’s behavioural susceptibility model of obesity (see; [90,91]). In Wardle’s model, food cue reactivity, which overlaps conceptually with what we term food-specific hunger, is exaggerated in people with obesity, with this having a genetic basis (e.g., [5,92]). The claim of exaggerated food cue reactivity in obesity relies upon two main strands of evidence: (1) differences in physiological markers between people who are lean and obese when exposed to food cues (e.g., enhanced salivation and other cephalic phase responses); and (2) differential regional brain activity between people who are lean and obese when exposed to food cues (e.g., [93,94,95]). However, evidence for differences in (1) and (2) are mixed and a recent systematic review and meta-analysis of the neuroimaging data found no weight-related differences [96]. While experimental studies examining the impact of food cue exposure on eating reliably find it promotes intake, surprisingly, this is not moderated by body weight as an exaggerated food cue reactivity account would expect [97]. Consequently, there remains much uncertainty about food cue reactivity’s causal role in obesity. 

One possible reason for this ambiguity may lie in how MTL impairment affects food-specific hunger. As described earlier, MTL impairment may exert a small negative effect on people’s capacity to retrieve episodic memories of eating palatable food. There may be two opposing forces at work on food-specific hunger in people with obesity. The first reflects genetic influences that enhance food cue reactivity and that are likely to be stronger in people with obesity. The second force, which stands in opposition to the first, reflects the impact of impaired MTL function on food-specific hunger, which may serve to weaken the capacity to vividly re-experience the sensory pleasure associated with a particular food, thereby reducing the appeal of a food cue (i.e., reducing hunger). As people differ in their genetic propensity for food cue reactivity (e.g., [5,92]) and probably also in their propensity to experience MTL-related damage, this may contribute to the observed heterogenous relationship between food cue reactivity and obesity. In other words, food-specific hunger may be impaired in some obese people, but less so in others.

As noted earlier, Schachter [89] and Wardle [92], both suggested that people with obesity had an impaired capacity to experience interoceptive states, including satiety. Early work in this area looked for relationships between adiposity and sensitivity to stomach contractions, but the results are inconclusive (see [98], for summary). A recent meta-analysis of the relationship between adiposity and general interoceptive ability [99], found only a very small (but significant) negative relationship (Cohen’s d = 0.1). While impaired interoceptive capacity may not be a general problem in obesity, it still may be for a subset of this population (e.g., [100]). A related consideration is internal sensitivity to the passage of time (i.e., one form of temporal cue). Although this was widely explored, this does appear to be impaired in people with obesity [101].

From the perspective of this manuscript, obesity should manifest a particular set of deficits relating to hunger and satiety, which should be a more exaggerated form of those predicted for a WS-diet, to the extent that they share some common cause in impaired MTL function. First, there should be blunting of general hunger, covering both interoceptive and temporal cues. Thus, when an individual with obesity senses one of their bodily hunger cues or notices that it is lunchtime, this should not generate much excitation transfer (notice that the deficit here is not in interoceptive ability [i.e., effectively perception], but in the learning and memory processes that follow). Consequently, any food cue that is encountered after experiencing a temporal or bodily hunger cue would not be augmented—that is made more appetising. This would mean that low-to-moderate-palatability foods such as fruit and vegetables would not have their expected consummatory pleasure augmented if observed with or shortly after an interoceptive or temporal hunger cue. Thus, people with obesity should have a reduced capacity for general hunger. Notice that this could coincide with an exaggerated food-specific hunger (i.e., food cue reactivity), in which case, the only foods that would excite hunger would be those of high palatability.

The second focus of dysfunction concerns satiety and memory inhibition. In just the same way that WS-diet-induced MTL impairment adversely affects memory inhibition and hence recalled food affect when sated (see Figure 1C), the same or a stronger version of this effect should be apparent in people with obesity. That is, food-specific hunger cues should still lead to the retrieval of positive food-related memories even when interoceptive cues for satiety are present. The consequence of this should be to increase the likelihood of eating when food-specific hunger cues are present, irrespective of satiety, meal recency, etc. While this was never directly tested, there is evidence for impaired memory inhibition on other non-food-related tasks in children and adolescents with obesity [102].

In summary, this suggests that some people with obesity may experience strong specific hunger for palatable foods, little general hunger, and little modulation of their food-specific hunger by satiety (i.e., impaired memory inhibition). This profile, which is of unknown prevalence, would be particularly disruptive, as the only cues that would motivate food consumption would be those for high-palatability foods, and this food-specific hunger would not be modulated by satiety. In addition, there would be little general hunger and hence little excitation transfer to increase the appeal of low-to-moderate-palatability foods. The appeal of these low-to-moderate-palatability foods would also be adversely impacted both by the effects of context (i.e., unprocessed foods appearing hedonically bland) and by the food memory landscape (i.e., the many adverts for palatable foods).

## 6. Anorexia Nervosa (AN)

Anorexia nervosa (AN) is a medical condition occurring in 0.1–3.6% of predominantly young women around menarche and is characterised by energy restriction resulting in abnormally low body weight, fear of weight gain, and disturbed body image. It has two major subtypes: a restricting form, with an earlier age of onset and better prognosis, and a binging/purging form with a poorer prognosis [103]. Many of the studies described below use samples containing both subtypes, and where a specific subtype is used, this is noted. 

For AN, during active weight loss (i.e., acute disease), neuroimaging indicates a general reduction in grey matter volume and cortical thinning—which is especially marked for several MTL structures (e.g., amygdala, hippocampus) as well as for the cerebellum, insular cortex, pre-frontal, and cingulate cortices [104,105,106]. These brain changes seemingly recover following weight regain [105]. Several studies examined reports of hunger and satiety in the active weight loss phase in people with restricting type AN. The consistent finding is of low pre-meal hunger and high pre-meal satiety, alongside little change in either hunger or satiety across the meal [107,108,109,110]. The most extensive investigations were undertaken by Halmi et al. [107,108] with relatively large clinical sample sizes (*n*’s > 30) and controls, and with comparisons of AN subtypes. In these studies, hunger and satiety were established before and after a liquid test meal, and there were few obvious differences between AN subtypes, with all exhibiting the pattern of hunger and satiety ratings described above. There is some evidence that reports of hunger/satiety normalise with treatment and weight regain, but this is not so well-established [110]. While these findings suggest that abnormal hunger and satiety may be a component of the active disease, there is no direct evidence connecting this to impairments in MTL function, nor is there any evidence to rule out other explanations (e.g., perhaps people with AN may deliberately under-report feeling hunger to explain why they will not eat; they may have physical complaints that preclude hunger: pain, nausea, dehydration, etc.). 

An additional observation concerns interoceptive states in AN. Interestingly, gut-related abnormalities—longer transit times, stomach dysrhythmia, delayed gastric emptying, and swallowing problems, are more frequent in AN populations than matched controls, and are sometimes evident before disease onset [111], which might be expected to affect the capacity to learn typical gut-related interoceptive cues. Moreover, interoceptive states are known to be abnormal in AN when measured by self-report on the Eating Disorders Inventory [112]. Experimental data suggest that while changes to bodily state, such as shifts in blood glucose or a filling stomach, can be detected by people with AN (e.g., [113]), they do not seem to have the same meaning as they do in healthy participants [114,115]. In Nakai’s [114] study for example, people with restricting AN subtype were given an insulin challenge, while their blood glucose was monitored and hunger ratings were obtained. While hunger in the healthy control group increased following insulin, the opposite occurred in the AN group, with a decrease in hunger, although a similar decrease in blood glucose was observed in both groups. Silverstone and Russell [115] noted that people with AN interpret their bodily sensations abnormally, suggesting that what may constitute an interoceptive hunger signal in a healthy person, does not do so in a person with AN. Whether this is a consequence of abnormal learning or of MTL damage, was also not evaluated. 

## 7. MTL Epilepsy

The commonest site of epileptogenic foci are structures within the medial temporal lobes and particularly in the hippocampus. Community prevalence of medial temporal lobe epilepsy is estimated to be around 0.2%, and conservatively, at least one half of these people have imaging-identifiable lesions termed hippocampal sclerosis (e.g., [116]). 

For MTL epilepsy, there are no specific studies examining hunger or satiety, and few on weight control. Interpretation of any potential findings is also complicated because of the confounding that arises from the drugs used to treat this condition. First, many of these drugs affect weight regulation (e.g., topiramate is now used as a weight loss agent; [117]). Second, many epilepsy drugs also actively prevent further MTL damage (this is discussed in the final part of the manuscript). Thus, examination of hunger and satiety in a medicated TLE epilepsy patient would not be straightforward to interpret even if it was studied. However, one group that is of interest are those patients who have drug-resistant MTL epilepsy. Drug resistance indicates a more severe presentation with greater associated MTL damage. Drug-resistant TLE epilepsy is now known to be linked to weight gain, suggesting an appetitive impairment [118], but exactly how this manifests is yet to be examined. Based on deficits identified following other forms of MTL damage, a hunger/satiety profile similar to that predicted for obese individuals would be expected, namely strong specific hunger for palatable foods, minimal general hunger, and impaired modulation of food-specific hunger by satiety.

## 8. Post-Traumatic Stress Disorder (PTSD)

Among US adults, PTSD has a lifetime prevalence of around 8% placing it as one of the commonest anxiety disorders; however, prevalence is much higher in certain populations with up to a quarter of US veterans having this condition [119]. Exposure to an event that poses a significant threat of death or injury and that produces a sense of fear, helplessness, or horror, can result in PTSD, which is characterised by recurrent flashbacks and avoidance of cues to the event. PTSD can run a chronic course and is costly to health, with very strong associations to substance abuse, depression, and increased risk for cardiovascular disease. 

There is consistent and strong evidence of MTL abnormality in PTSD, especially relating to hippocampal and amygdala function as established by neuropsychological testing and structural neuroimaging (e.g., [120]). While specific studies of hunger and satiety were not undertaken, there is considerable evidence for abnormal patterns of eating behaviour in people with PTSD. First, rates of obesity are much higher in people with PTSD relative to control samples (e.g., [121]). Second, incidence of eating disorders, most notably binge eating disorder (BED), are much higher than in control populations, and PTSD was identified as a specific risk factor for BED (e.g., [122]). As the name implies, BED is characterised by binging (eating a large amount of food in a short period of time with a feeling of loss of control), and it seems plausible that disturbances of both hunger and satiety could contribute to disease onset and maintenance. While this may be the case, it is important to note that neuroimaging studies tended to identify a different set of brain structures as being abnormal in BED—notably the striatum, insula, and prefrontal cortex (e.g., [123]). However, as the disease name suggests (post-traumatic stress disorder) abnormalities in the stress response, with their very well-established linkage to both amygdala and hippocampal dysfunction, and appetitive abnormalities [124], do suggest an MTL-based cause.

## 9. Implications for Intervention

One conclusion to emerge from the perspective adopted here is that improving appetitive regulation via hunger and satiety will have to involve ameliorating or repairing MTL-based learning and memory systems, which we argued are critical for supporting hunger and satiety. One approach would be to use drugs that improve MTL function, which would in turn normalise hunger and satiety, and so contribute to stabilising body weight. There are already several drugs that are known to benefit the MTL, and especially the hippocampal formation [125]. These include: (1) drugs used in Alzheimer’s disease to enhance cognition, namely donepezil and memantine; (2) medications used in epilepsy, notably phenytoin and sodium valproate; (3) certain antidepressants (fluoxetine, tianeptine); (4) other pharmaceuticals such as mifepristone and lithium [125]; and (5) certain type 2 diabetes medications (i.e., glucagon-like peptide [GLP-1] agonists; [126]). Knowledge about the specific effects of these drugs on hunger and satiety, and more broadly weight gain/loss, is incomplete, but there is evidence that all of these drugs impact ingestive behaviour. It is largely unknown if their biological effects on the MTL (and its psychological consequences for memory function) are correlated with any effect that they may have on ingestive behaviour. 

Some of the drugs identified above were linked with weight gain and an increase in appetite (lithium, sodium valproate; e.g., [127]), some with weight loss and appetite suppression (memantine, donepezil, fluoxetine, phenytoin, and mifepristone; e.g., [128,129]) and some with both (tianeptine). The GLP-1 agonists are particularly intriguing, as they favourably impact both hippocampal function and appetite-related processes (e.g., [126,130]). Recent animal work suggests they can facilitate memory inhibition, and so enhance satiety and suppress hunger [131]. As there is somewhat less recognition outside of psychology for the role of MTL-related memory processing in appetite regulation, there is potential for both novel drug treatments (e.g., memantine was successfully trialed for binge eating disorder; [132]), and critically, for exploring the combination of drug treatments with psychological interventions to enhance the functional utility of hunger and satiety.

Indeed, there is a lot of interest in trying to improve people’s ability to use hunger and satiety to bring their food intake back into line with their energy needs (e.g., [133,134]). However, this research is not always guided by an overarching idea as to how hunger and satiety might operate. Moreover, in some cases, especially for the conceptualisation of interoceptive hunger cues, thinking was driven by a homeostatic perspective (i.e., the presence of this internal cue indicates an energy need, hence one should eat). This belief about interoceptive hunger cues dominates many people’s idea of how hunger is caused [44,135]). While certain interoceptive states may be good indicators of when food will be rewarding to eat, they may not always be good indicators of the body’s need for energy (e.g., low mood has a particular feeling which indicates hunger in some people, but low mood can occur independent of energy needs).

A further consideration is that some people may not be able to utilise alimentary interoceptive cues for hunger and/or satiety, as they may never have learned them as children. Due to the heritable nature of weight gain propensity in an obesogenic environment [5], many people who are currently obese probably had/have at least one or both parents who were obese when they were a child. In this case, if obesity is related to a more limited capacity to utilise alimentary interoceptive hunger and satiety cues, they may not have been able to teach their child about them. It is then possible that children of obese parents never learned (or learned less) about alimentary interoceptive signals, leaving them more reliant on other cues to regulate feeding. These other cues could be food-related stimuli (i.e., environmental events), mood states, or temporal cues, such as lunchtime.

Several attempts were made to teach children and adults to attend to their interoceptive hunger and satiety signals (e.g., appetite awareness training; [136]). This can vary between highly specific training, where people learn to link hunger to momentary decreases in blood glucose, and approaches in which people are asked to judge how hungry (or full) they feel, and then decide if this feeling is sufficiently strong (or weak) to trigger or stop eating. In a clinical context, some people report that these interventions can be helpful in guiding them to avoid reliance on environmental food cues or (typically) low mood as a source of hunger. However, it is unclear why some people seem to benefit more from training than others and how they might be identified (see [100]). It would also be interesting to undertake what may be a more naturalistic form of training mimicking what may be occurring during childhood, with the selection of a particular interoceptive cue, and then eating various foods in response to this cue. The main advantage of doing this may be to facilitate consumption of low-to-moderate-palatability foods such as fruits and vegetables via excitation transfer, which might assist a person to switch to a more healthful diet by making it more appealing (e.g., Mediterranean, MIND or similar). Such diets seem to offer additional benefits to cognition and MTL function (e.g., [137,138]).

A further approach would be to consider training with external time cues, which arguably can come to function in a similar manner to interoceptive hunger cues. The interesting aspect of focusing on this class of general hunger cue, is that using clock time does not rely on any problem/ambiguity in perceiving interoceptive states. The person would then have to learn to eat at particular times each day, so that these times comes to serve as general hunger cues, generating excitation transfer to any food cue encountered around that time. There may be additional strategies that could be employed to maximise excitation transfer. If excitation transfer is the average of many different positive food experiences, then the average could be enhanced by intermittent inclusion of high-palatability foods within the context of a meal. This would have the effect of increasing the average degree of excitation transfer, so that even if low-to-moderate-palatability foods are normally consumed, they would benefit from greater excitation transfer. Practically, this might mean routinely eating at certain times of day, and ensuring that every few days at each mealtime, something particularly delicious is consumed. These habits are then built deliberately so that an external cue that the person is readily able to perceive—clock time—comes to serve as a hunger cue, one that then generates excitation transfer to food memories, from for example, thinking about what to have for lunch. 

The capacity to utilise interoceptive satiety cues may also be problematic in some people. Any approach to better utilising interoceptive satiety cues may depend on improved MTL function, something that would possible not occur until diet change, weight loss, or medication were initiated. Indeed, perhaps the broader point here is that training or re-training of alimentary interoceptive cues may require a fully functional MTL system, something that is unlikely in many people wanting to undertake such training. It is for this reason that the development of alternate methods (e.g., drugs, focus on clock time) might be particularly beneficial. Finally, another approach which may supplement satiety in regulating intake, builds upon the work of Higgs and colleagues, and may also rely upon an intact MTL system. It involves having people explicitly recall their prior eating bout, which under experimental conditions, can reduce subsequent food intake (e.g., [139,140]). This approach was also suggested to draw upon the hippocampus’s ability to engage in inhibition (e.g., [69]).

## 10. Mechanism 

Not surprisingly given the findings described here, there is now an extensive literature identifying the neural basis for the medial temporal lobes, and especially the hippocampus’s role, in appetitive processing (e.g., [67,141]). The hippocampus has the necessary links to other brain regions involved in controlling food intake, notably the hypothalamus [142]. It also receives inputs from the vagus, carrying ingestion-related information, as well as having receptors for many of the key hormones involved in appetite regulation (notably insulin, ghrelin, and leptin; e.g., [143]). Many of these hormones also serve a dual function in both facilitating learning and memory, as well as being food intake regulators [67,141]. The hippocampus is then well situated to evaluate whether an internal or external cue for food is going to be rewarding or not to consume.

As we noted earlier, the hippocampus appears to be particularly vulnerable to insult (e.g., [24,25,26,27,28]). There is a great diversity of cause amongst these insults. A WS-diet involves exposure to saturated fat, added sugar, and salt. Obesity and its comorbidities affect glucoregulation and other metabolic systems. Epilepsy involves abnormal electrical activity in the brain. PTSD results from exposure to traumatic experience, and AN from starvation and chronic stress. Yet, in all of these cases, there is resultant hippocampal dysfunction, as there also is from many other agents and causes not discussed here, such as viral infections (HSV-1), ischemia/hypoxia, and traumatic brain injury [25]. 

One possible source of this vulnerability concerns the nature of the hippocampus itself. It is neurochemically diverse (e.g., [144]), with glucocorticoid, insulin, leptin, ghrelin, and incretin receptors, which in normal operation are neuroprotective, but when dysregulated, as with chronic stress (e.g., PTSD and AN, with excessive glucocorticoid exposure) or metabolic disturbances (e.g., insulin or leptin in obesity for example), can dispose to injury. In addition, it has a large diversity of cell types and complex circuits (e.g., [145]). This increases the possibility of damage from diverse sources, with, for example type 2 diabetes differentially affecting CA1 and the subiculum, while CA3 and CA1 appear susceptible to the effects of medial temporal lobe epilepsy [25]. In addition, the hippocampus is particularly well connected to other brain regions, making it potentially vulnerable to damage in remote locations via transneuronal degeneration [146].

Many of the conditions discussed in this manuscript share three common pathophysiologies [25]. The first is increased oxidative stress (e.g., from a WS-diet), with greater production and/or decreased removal of free radicals. The CA1 region of the hippocampus appears to be particularly susceptible to oxidative stress [147]. The second is neuroinflammation (e.g., from obesity). This typically starts with a peripheral source (e.g., from a leaky gut or from WS-diet components such as saturated fat) that result in the relaease of pro-inflammatory cytokines. These pass through the blood–brain barrier (BBB), resulting in the activation of the microglia, and producing a neuroinflammatory response. The hippocampus has an exceptionally dense array of microglia, and there is evidence from animal models that peripheral inflammation results in a particularly rapid activation of this microglia array in the hippocampus [148]. The third is glutamate excitotoxicity, with excess glutamate producing an over-stimulation of glutamate receptors, neurotoxicity, and neuronal death. The CA1 region of the hippocampus has an unusually high density of glutamate receptors and so may be particularly susceptible to this [149].

A final consideration, and one that we examined elsewhere in greater detail (see [25]) is the possibility that all of the agents/insults considered here may exert a common effect via their ability to disrupt the BBB. The BBB is heterogenous, and some brain areas have a BBB that is much more susceptible to disruption than others—althought the causes of this differential susceptibility are poorly understood. The hippocampal region is one such area, with for example, changes to blood glucose levels, insulin, and the electrical activity of seizures, all known to affect its integrity [25]. Alterations in the BBB’s permeability could allow toxins to enter the brain, promoting neuroinflammation and neuro-metabolic disturbances, and dysregulating hippocampal function.

Beyond the hippocampus, other MTL structures are also important components of the processes described in this manuscript. While the amygdala was studied in animals regarding its role in simple Pavlovian cue-food learning, other structures linked to declarative learning and memory received far less attention. Amongst these, the ATL/MTL structures supporting semantic memory may be critical for assigning meaning to interoceptive and temporal cues, but work in this area in humans is limited to a few neuropsychological studies (i.e., [56,57]), with no neuroimaging investigations at all. More generally, the integration of episodic and semantic memory systems, or as here, the integration of general and specific hunger through excitation transfer, is a further area that received little investigation, either from neuropsychology or neuroimaging. Finally, whether these brain areas also share some or all of the vulnerabilities to damage outlined above for the hippocampus is not fully understood.

## 11. Future Research Directions

In focusing here on a psychological approach to hunger, the question arises as to how this integrates with the biological approach to hunger that is a cornerstone of research in this field. A biological approach is one based upon understanding the physiological changes that accompany all aspects of energy utilization and storage, and the processes of digestion and absorption, which are presumed to regulate food intake and cessation (e.g., [150,151,152]). The fundamental assumption made in this manuscript, is that the conscious outputs of all of these biological systems only acquire meaning (i.e., food will be rewarding [or not] to eat now) due to declarative learning and memory processes. Exploration of this assumption is currently limited (but see [11]), but it seems critical to explore for two reasons. First, as we described earlier, many common diseases and conditions impair the learning and memory processes involved in supporting hunger and satiety. Thus hunger and satiety are vulnerable processes because of their neural substrate, which lends great practical significance to their study. Second, even key aspects of ingestive physiology do not seem to yield the type of control over appetite that their apparent physiological significance would suggest. The most telling example concerns Agouti-related peptide (AgRp) produced in AgRp/NPY neurons of the arcuate nucleus of the hypothalamus. AgRp has a very well-established orexigenic effect if these neurons are stimulated (e.g., [152]). Elimination of AgRp in adult mice is catastrophic, as the mice starve to death, presenting a seemingly stark example of the importance of these biological signals for motivating food intake: hunger. However, this conclusion is both incomplete and misleading. Knockout animals born with no AgRp (or NPY or both) are essentially developmentally normal, feeding and gaining weight as control animals [153]. The implication is that the mouse learns that certain interoceptive cues (i.e., what it feels like following neuropeptide release) signals that food will be rewarding to eat now. As there are many potential biological cues that can be learnt, reliance on any particular one is not critical, but once learned, the mouse at least may not be readily able to learn others—at least in the time available before it starves to death. If we are to understand how to better regulate hunger and satiety through physiological means (e.g., drugs, etc.), it is necessary to understand how these biological processes acquire meaning through learning and memory. This is arguably the most important future direction. 

## 12. Conclusions

Learning and memory processes, particularly those utilising MTL substrates, are receiving increasing attention for their contribution to the control of ingestive behaviour (e.g., [6,64,67,141,154]). Damage to MTL structures can occur in many disease states/conditions, some closely linked to ingestive behaviour (e.g., WS-diet, obesity, anorexia nervosa) and others not (e.g., temporal lobe epilepsy, PTSD). While much remains to be tested to ascertain the full extent of hunger and satiety-related impairments across these diverse conditions, there is evidence, stronger in some cases weaker in others, that supports a linkage between MTL integrity, declarative learning and memory function, and hunger and satiety. These findings suggest that focusing on strategies such as pharmacological approaches that aid MTL recovery and psychological approaches to hunger/satiety training, and critically their combination, may provide improved methods for helping people better regulate their appetite.

## Figures and Tables

**Figure 1 nutrients-16-03013-f001:**
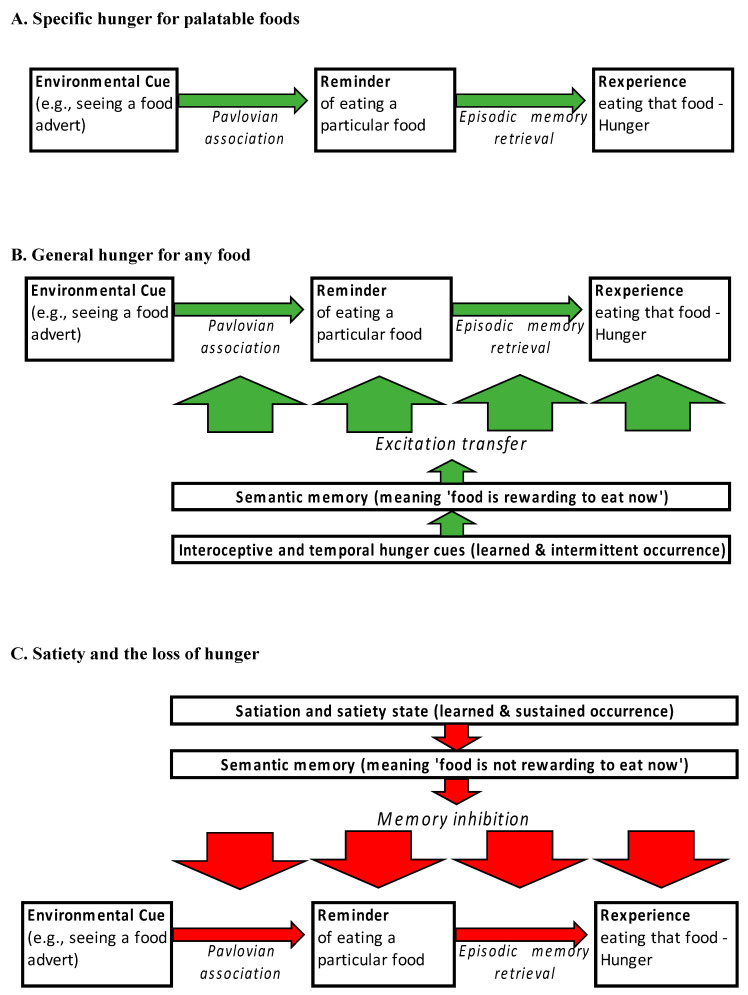
Declarative memory models of specific and general hunger and satiety.

**Figure 2 nutrients-16-03013-f002:**
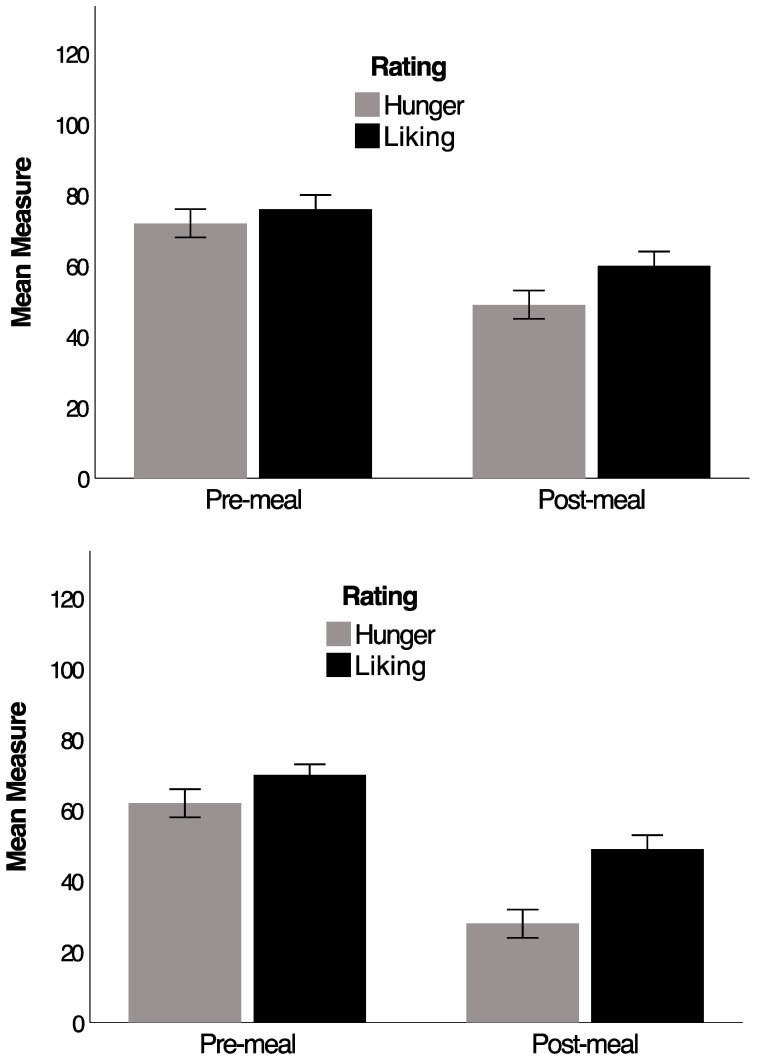
Upper panel: Participants with a Western-style dietary pattern, with their mean ratings (and standard error) for palatable snack foods for specific hunger (on looking at the food) and liking (on tasting the food) before and after a meal. Lower panel, the same, for participants with a healthier dietary pattern. Data adapted from [66].

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
