# Peer review of "Hunger, Satiety, and Their Vulnerabilities"

_nutrients, 2024, doi:10.3390/nu16173013_

Round 1

Reviewer 1 Report

Comments and Suggestions for Authors

The paper presents a psychological approach to hunger and fullness. There are plenty of general statements and a history of research. Consequently, the paper is much different from those presented in medical journals. The authors did not try to combine current neurophysiological knowledge with psychological findings.

It is hard to accept that hunger is 'psychological state' - in physiology sensation.  Satiety is the opposite, and this term is used - not 'fullness'.

The authors included in the title - vulnerability of hunger and fullness but is poorly addressed in the paper's content. 

The lecture of the paper is not very useful to the potential reader. 

There is nothing about 'food craving'.

Author Response

>> We thank the reviewer for reading and commenting on our manuscript.

The paper presents a psychological approach to hunger and fullness. There are plenty of general statements and a history of research. Consequently, the paper is much different from those presented in medical journals. The authors did not try to combine current neurophysiological knowledge with psychological findings.

>> It is correct that our focus was more on the psychological findings, as this was the main thrust of the paper.  We do agree that more anchoring into the physiology and neurophysiology would be advantageous, and to this end we added two sections, one indicating that the MTL has the capabilities to undertake the type of processes we envisage (please see Section 10), and one outlining why the major scientific question is now how indeed we reconcile psychological and physiological accounts of hunger and satiety (please see Section 11).

It is hard to accept that hunger is 'psychological state' - in physiology sensation. 

>> The new Section 11 attempts to address this issue more directly.

Satiety is the opposite, and this term is used - not 'fullness'.

>> Thank you for pointing this out.  We have corrected this throughout the manuscript.

The authors included in the title - vulnerability of hunger and fullness but is poorly addressed in the paper's content. 

>> We have now detailed more clearly what we mean by vulnerabilities, at the start of the section dealing with them (please see Section 3) and later in Section 11 (line 766 onwards).

The lecture of the paper is not very useful to the potential reader. 

>> Hopefully we have made it clearer, why it is important to consider a psychological perspective.

There is nothing about 'food craving'.

>> This is now flagged, as it can be seen as an exaggerated form of hunger (please see line 104).

Reviewer 2 Report

Comments and Suggestions for Authors
  1. Methodological Clarity: While the manuscript provides a strong theoretical basis, there is a need for more detailed discussion on the methodologies employed in key studies cited. Including more specifics about study designs, sample sizes, and statistical analyses would enhance the reader's understanding and the manuscript’s rigor.
  2. Future Research Directions: The manuscript would benefit from a more explicit section outlining potential future research directions. While some gaps are noted, a dedicated section on this would provide clearer guidance for subsequent studies.
  3. Terminology Consistency: The manuscript occasionally uses specialized terminology without prior definition. For instance, terms like "memory inhibition" and "excitation transfer" could be briefly defined when first introduced to aid readers who may not be familiar with these concepts.
  4. Balance of Conditions: There is a noticeable emphasis on the Western-style diet and obesity, whereas conditions like anorexia nervosa, temporal lobe epilepsy, and PTSD receive comparatively less detailed treatment. Providing a more balanced discussion across all conditions examined would strengthen the manuscript.
  5. Mechanistic Insights: While the manuscript adeptly links MTL function to hunger and fullness, there is room for more mechanistic insights into how specific neural pathways and neurotransmitters within the MTL contribute to these processes. Including more recent findings from neuroimaging and neurophysiological studies could deepen the discussion.

Author Response

>> We thank R2 for reading and commenting on our manuscript.

  1. Methodological Clarity: While the manuscript provides a strong theoretical basis, there is a need for more detailed discussion on the methodologies employed in key studies cited. Including more specifics about study designs, sample sizes, and statistical analyses would enhance the reader's understanding and the manuscript’s rigor.

>> There are several key studies, where we have now increased the detail about the design, manipulations, sample size, measures, approach and statistics etc – although the varied nature of the studies means that certain of these details are more relevant to some studies than to others.  The ones we focussed on were the original Hebben et al., 1985 study on HM (ln 74 and ln82 onwards),  Francis & Stevenson, 2013 (ln226), Attuquayefio et al., 2017 (ln241), Attuquaefio et al 2016 (ln289), Stevenson et al., 2020 (ln 296), details around the affective discrepancy experiments (ln272), the Halmi et al studies (ln522) and Nakai study (ln544).

  1. Future Research Directions: The manuscript would benefit from a more explicit section outlining potential future research directions. While some gaps are noted, a dedicated section on this would provide clearer guidance for subsequent studies.

>> We have included a new Section 11 to address this issue, but noting as the Reviewer stated that more specific issues are dealt with in the body of the manuscript.  The new section focusses on what we feel is both the major theoretical question and the one that has been least adequately addressed in the literature.

  1. Terminology Consistency: The manuscript occasionally uses specialized terminology without prior definition. For instance, terms like "memory inhibition" and "excitation transfer" could be briefly defined when first introduced to aid readers who may not be familiar with these concepts.

>> These terms have now been defined and our apologies for this oversight before (please see ln151 and ln162).

  1. Balance of Conditions: There is a noticeable emphasis on the Western-style diet and obesity, whereas conditions like anorexia nervosa, temporal lobe epilepsy, and PTSD receive comparatively less detailed treatment. Providing a more balanced discussion across all conditions examined would strengthen the manuscript.

>> We have gone some way to remedy this by expanding these sections (please see Sections 6, 7 and 8), but we are limited by the fact that the literature on WS-diet and Obesity has often directly addressed the issues the MS is focussing on, while those for AN, TLE and PTSD have not been so addressed – hence the literature here is much less well developed.

  1. Mechanistic Insights: While the manuscript adeptly links MTL function to hunger and fullness, there is room for more mechanistic insights into how specific neural pathways and neurotransmitters within the MTL contribute to these processes. Including more recent findings from neuroimaging and neurophysiological studies could deepen the discussion.

>> A section addressing this has now been included (please see Section 10).  As we note in that section, there are several excellent papers that deal with these mechanistic issues, notably recent ones cited by Kanoski et al and by Parent et al. (both cited).